# Insights from Population Genomics to Enhance and Sustain Biological Control of Insect Pests

**DOI:** 10.3390/insects11080462

**Published:** 2020-07-22

**Authors:** Arun Sethuraman, Fredric J. Janzen, David W. Weisrock, John J. Obrycki

**Affiliations:** 1Department of Biological Sciences, California State University San Marcos, San Marcos, CA 92096, USA; 2Department of Ecology, Evolution, & Organismal Biology, Iowa State University, Ames, IA 50010, USA; fjanzen@iastate.edu; 3Kellogg Biological Station, Michigan State University, Hickory Corners, MI 49060, USA; 4Department of Biology, University of Kentucky, Lexington, KY 40506, USA; david.weisrock@uky.edu; 5Department of Entomology, University of Kentucky, Lexington, KY 40506, USA; john.obrycki@uky.edu

**Keywords:** population genomics, biological control, demographic models, pest management

## Abstract

Biological control—the use of organisms (e.g., nematodes, arthropods, bacteria, fungi, viruses) for the suppression of insect pest species—is a well-established, ecologically sound and economically profitable tactic for crop protection. This approach has served as a sustainable solution for many insect pest problems for over a century in North America. However, all pest management tactics have associated risks. Specifically, the ecological non-target effects of biological control have been examined in numerous systems. In contrast, the need to understand the short- and long-term evolutionary consequences of human-mediated manipulation of biological control organisms for importation, augmentation and conservation biological control has only recently been acknowledged. Particularly, population genomics presents exceptional opportunities to study adaptive evolution and invasiveness of pests and biological control organisms. Population genomics also provides insights into (1) long-term biological consequences of releases, (2) the ecological success and sustainability of this pest management tactic and (3) non-target effects on native species, populations and ecosystems. Recent advances in genomic sequencing technology and model-based statistical methods to analyze population-scale genomic data provide a much needed impetus for biological control programs to benefit by incorporating a consideration of evolutionary consequences. Here, we review current technology and methods in population genomics and their applications to biological control and include basic guidelines for biological control researchers for implementing genomic technology and statistical modeling.

## 1. Introduction

Biological control—the use of natural enemies or biological control organisms such as terrestrial arthropods, microorganisms and invertebrates (e.g., entomophagous nematodes) to suppress populations of agricultural pests—has been a successful pest management tactic for over a century [1,2,3]. Motivated by the abundance of naturally occurring predator-prey and parasitoid-host species interactions, biological control provides benefits for pest suppression. Such benefits include the potential for long-term pest suppression and increased environmental and human safety, in comparison to the use of chemical insecticides [4]. Examples of highly successful and sustainable biological control include programs for the ash whitefly, cereal leaf beetle, alfalfa weevil and the cassava mealybug [5,6,7,8] and for additional examples see References [3,9].

However, human-mediated release of biological control organisms may have short- and long-term consequences for the evolution of (a) prey/hosts (also called ‘target’ effects), as well as (b) released populations of biological control organisms, (c) native (resident) populations, that may compete with released biological control organisms and (d) associated endosymbiont/microbial diversity (collectively termed as ‘non-target’ effects) which could detrimentally affect other species that interact with the biological control organisms [10]. At the ecological level, both target and non-target effects of biological control have been studied broadly in the context of efficacy and efficiency of control strategies [3]. Such examples include species interactions and resource competition [11], host-pathogen interactions and interactions of biological control organisms with endosymbionts and transgenic host plants [12]. Research to improve biological control programs, even to push a 10% increase in success of importation and augmentation, continues to be a challenge [13].

With the advent of modern sequencing technologies and statistical methods to analyze large-scale genetic data, agriculturalists and geneticists are increasingly applying population genomics as a means to enhance our understanding of the evolution of biological control organisms and insect pests [14,15,16]. Such a strategy is mindful of not just the immediate consequences of introducing biological control organisms for pest suppression but of long-term evolutionary trajectories of both the pest and biological control species [17]. Genomic data can offer uniquely valuable insights into changes in population size, natural and artificial selection, migration or admixture, inbreeding and even co-evolution of biological control organisms and their pest targets. Population genomics hence provides an efficient means of monitoring these important factors for success of biological control programs. Studying biological control organisms also presents a unique and controlled opportunity to address fundamental questions about adaptive evolution, invasiveness and co-evolution.

This review focuses on a range of fundamental issues that have been addressed using population genomics in general but have yet to be applied to gain a better understanding of biological control. We first summarize different methods of biological control and population genetic models that can be used to describe them. We then focus on four core issues involving population genomics and biological control—(1) population size change, (2) natural selection and adaptive evolution in novel environments, (3) gene flow and (4) inbreeding. Finally, we provide recommendations and an outline of suggested steps (a ‘pipeline’) for researchers to facilitate use of available genomics methods to assess biological control. The emphasis of this review is on entomophagous species, that is, predators and parasitoids that attack insect pests.

## 2. Application of Population Genomic Models to Biological Control

Biological control of insect pests can be classified broadly into three methods, based on the mode(s) of manipulation of biological control organisms—importation, augmentation and conservation. In this review, we discuss importation and augmentation, the two methods in which arthropod biological control organisms are released into the environment. Most introduction histories of entomophagous species are complex sequences of demographic events. These sequences of events in turn determine current genomic diversity, population densities, sustainability and thus success of biological control. Also, although detailed historical introduction records have been maintained for many species of biological control organisms [18]—specific example, the predatory lady beetle, *Coccinella septempunctata* [19]—the quality of data for many species is highly variable. This is especially true for some species of insect predators that have become invasive [20,21]. However, their post-importation and augmentation history can be inferred using population genetic models. These models represent how populations grow or decline in numbers, evolve, exchange genes and diverge. Here we discuss biological control scenarios and population genetic models that can be used to infer post-introductory evolutionary histories.

**(a) Importation biological control** is defined as the introduction of biological control organisms in a single or repeated pulse(s) into a previously unoccupied environment [4]. Examples of successful importation include the vedalia beetle, *Rodolia cardinalis* [22] and many species of insect parasitoids [4,6]. Importation can be modeled using a “serial-founder” model [23], Figure 1A).

Serial founding of biological control organisms can occur naturally due to invasiveness or be anthropogenically mediated due to importation. Examples of serially founded biological control organism populations include an egg parasitoid (*Trissolcus japonicus*) of an introduced insect pest species, *Halyomorpha halys* [24], the Harlequin lady beetle, *Harmonia axyridis* [20,25] and the seven-spotted lady beetle, *Coccinella septempunctata* [19]. Serial founder models allow the estimation of numerous parameters, including times of serial founding of each population, genetic diversity and effective population sizes of the source and serially founded populations. Effective population sizes are different from census sizes, being more informative of the degree of genetic diversity within imported populations (see Box 1). Comparing effective population sizes of imported populations thus aids in understanding the degree of random genetic drift versus natural selection in driving their evolutionary dynamics. For example, Calfee et al. [26] compared genetic diversity of Africanized honey bees, *Apis mellifera scutellata*, in hybrid zones in North and South America and found no significant reduction in genetic diversity due to bottlenecks and rapid expansion. They combine these findings with a study of differential fitness, showing that natural selection has played a role in maintaining high genetic diversity in hybrid bees.

Serial founding can also incorporate gene flow between one or more founded populations to estimate migration rates and admixture parameters (see Box 1). This model further allows the estimation of “bridgehead” effects [20], which often lead to successful invasion and establishment of imported organisms in new environments.

Box 1Definitions of population genomic terms used in this article.Effective Population Size (Ne): The size of the population that is evolving neutrally due to random genetic drift. In a randomly mating population of constant size and in the absence of natural selection, this Ne should be equivalent to the census size, Nc. The Ne of a population is often approximated as a measure of its genetic diversity.Census Population Size (Nc): The number of individuals in a population of a species. Changes in the census size (e.g., due to competition from congenics, insecticide use) will also affect the rate of evolution by genetic drift and therefore the population’s effective population size, Ne. Nc is difficult to measure in nature, especially in natural enemies.Natural Selection: Changes in allele frequencies in a population due to differential fitness of alleles or combinations of alleles.Genetic Drift: Fluctuation in allele frequencies in a population due to random sampling of alleles from one generation to the next.Bottleneck: Decrease in the census size, Nc of a population, owing to importation or augmentation, leading to a decrease in its effective population size, Ne.Genetic Diversity: The diversity of alleles across genomic loci in a population (allelic richness) or the average heterozygosity across genomic loci. Genetic diversity of a population is directly affected by is Nc (and therefore Ne), mating processes (random versus non-random mating), geographical population structure and natural selection.Hybrid Vigor: Increased fitness of hybrid strains. In natural enemies, this could be quantified as increased fecundity, mating success, range expansion and invasiveness, competition success, resource utility.Deleterious Mutations: Alleles that confer lower absolute fitness and thereby lower relative fitness of genotypes that carry this allele in a population.Adaptation: Survival, reproduction and viability of heritable advantageous traits due to natural selection.Meiotic Recombination: Exchange of genetic material between maternal and paternal chromosomes during meiosis. Recombination landscape is affected by genetic drift and natural selection.Sexual Selection: Pre-mating barrier to gene flow, owing to differential mate choice. In arthropods, this could include wing or elytral patterning, chemical cues, vocalizations and size variation.Inbreeding: Non-random mating between close relatives within a population. Inbreeding could be opportunistic (due to geography, leading to the formation of structured populations) or due to sexual selection.Inbreeding Depression: Accumulation of deleterious mutations in inbred populations, leading to decreased fitness.Migration/Gene Flow/Admixture/Introgression: Physical movement and reproduction (therefore recombination) of migrant individuals from one population into another.Genetic Linkage: Co-inheritance of collinear segments of DNA owing to reduced recombination between them.Linked Selection: Co-inheritance of non-recombinant segments of DNA due to natural selection on a linked genetic locus.Genetic Hitchhiking: Process of co-inheritance of variants in non-recombinant segments of DNA due to positive natural selection on a linked genetic locus.Selective Sweep: Pattern of reduced genetic diversity in non-recombinant segments due to genetic hitchhiking.Quantitative Trait Loci: Genomic loci that control variability in quantitative phenotypes.Epistasis: Interaction across variants at different genomic loci, contributing towards variability in a trait.Sequencing Depth/Coverage: The average number of times every single nucleotide has been sequenced.Sequencing Read: A contiguous piece of DNA that is obtained from a sequencer, that have to be assembled to form contigs or often chromosome-size scaffolds.SNP’s: Single Nucleotide Polymorphisms - variants at a single nucleotide locus.

**(b) Augmentation biological control** embodies biological control organisms that were originally imported but failed to persist in their new environment and have their populations augmented through repeated releases, typically annually [27]. Examples of augmented biological control organisms include the greenhouse whitefly parasitoid, *Encasia formosa* and egg parasitoids in the genus *Trichogramma* [6,28], the mealybug destroyer, *Cryptolaemus montrouzieri* and over 230 commercially available arthropod species [29,30]. Arthropod biological control organisms from a stock population (often purchased *en masse*) can also be repeatedly introduced into an environment where they have already been established (Figure 1B) and can be modeled using a “source-sink” model. Under a source-sink model, demographic parameters such as effective population sizes of the founding source population and the recipient introduced populations and continued rates of unidirectional migration from the source to the sink population (in number or proportion of individuals per generation), can be estimated.

Population genetic models can describe aspects of biological control:

(a) Successful biological control programs can result in the establishment of introduced populations over a broad geographic range, sometimes through non-anthropogenic assisted range expansions. Examples of this process have been noted in the literature, including parasitoid Aphelinidae and Braconidae hymenopterans [31,32], the flower head weevil, *Rhinocyllus conicus* [33] and numerous invasive species (summarized in Reference [34]). This scenario can be modeled using an isolation by distance framework ([35,36], Figure 1C). Under this model, gene flow restricted to geographically proximal populations leads to increased genetic differentiation across the range of the introduced species (Figure 1C). Recent advances in utilizing genomic surveys to inform isolation by distance [37] could potentially be applied to long-range dispersal of organisms to infer fine-scale patterns of range expansions.

(b) Introduced populations of biological control organisms are often small. Thus their successful establishment depends on numerous factors, including adaptability to local environments, availability of hosts/prey and competitors. Modeling effective population size declines are thus informative of changes in genomic diversity in introduced populations and of potential utility in conservation biological control. Alternatively, unsuccessful introductions summarized in References [33,38], are also characterized by declining population sizes. Population size declines are often modeled using a bottleneck model for inbred, small populations [39,40], Figure 1D. Models incorporating population size change can estimate population growth or decline rates, along with effective population sizes of founder and introduced populations of biological control organisms. These factors can be used in tracing evolutionary trajectories and effectiveness of biological control (see discussion).

Importantly, numerous statistical methods use population genomic data to rigorously identify the best-fitting demographic model for a particular biological control system (see Table 1). Furthermore, these methods allow for the estimation of evolutionary parameters of specific interest to biological control (population size, rate of growth or decline, migration, etc.).

## 3. Genomic Signatures during Biological Control

Post-introductory demographic history of biological control organisms can be complex to model but can be characterized by estimating four major “parameters” of populations using genomic data—population size change, adaptation, admixture or migration and inbreeding [17], (Box 1). Here we provide an overview of these parameters and discuss how they can be estimated from genomic data derived from organisms released for biological control.

### 3.1. Population Size Change

Bottlenecks and change in effective population sizes both influence genomic diversity of species. Species utilized for biological control are subject to both these processes, depending on their natural history and interactions. Newly introduced populations of biological control organisms often undergo bottlenecks, where a relatively small sample of founder individuals from a larger population is introduced into a novel environment [17,75,76,77,78].

Conversely, population size growth can be enhanced in introduced populations via “invasiveness” or the uncontrolled growth of a population in a non-native (introduced) environment (e.g., *Harmonia axyridis*—[79]. Invasiveness of biological control organisms could be primarily due to plastic phenotypic response to changing environments [80], hybrid vigor [26,81] or rapid life-history evolution [82]. Expanding (and invading) populations evolve faster, owing to increased efficacy of selection in purging deleterious mutations and fixing advantageous ones, compared to declining or bottlenecked populations [83].

Inferring effective population sizes and changes serves as a primary indicator of population genomic processes affecting the ecological success of biological control (i.e., establishment of the biological control organisms followed by a reduction in the pest population density) and provides a much more informative alternative to otherwise detailed and labor intensive census size estimation. Applied in combination with other population genomics statistics, effective population size estimation is a means to building hypotheses to explain the success or failure of biological control programs (see Table 2).

### 3.2. Natural Selection and Evolution

Populations of biological control organisms in new environments, apart from undergoing population size change, are also subject to adaptive evolution in response to selection. Broadly, selection nudges populations towards fitness peaks [84].

The genetics of adaptive evolution in introduced and invasive species have been studied extensively but not in the context of biological control [21,85,86,87,88]. Numerous cases of failed introductions of biological control organisms have been noted, however, presumably owing to differential fitness [75,86,89], strong directional selection due to insecticide use [90] and sexual selection and the ‘Goldilocks principle’ [91] or adaptive evolution of traits to a selective optimum in response to environmental selection. Other factors that contribute to the success of biological control by influencing the rate of adaptive evolution of introduced individuals to the new environment include linked selection and divergence hitchhiking [92,93], migration and admixture [26,94] and inbreeding [95,96].

Multiple introductions of the same species, including populations from different geographic sources, can play a prominent role in local adaptation, invasiveness and boosting genomic diversity in populations of biological control organisms. Biological control has the distinction of having extensive introduction records over recent time scales [18,19], thus quantifying genomic variation of imported or augmented biological control organisms allows researchers and biological control administrators to study, with temporal validation, their adaptive potentials to new environments. Of particular interest are quantitative trait loci (QTLs) that contribute directly to adaptive evolution of biological control organisms in new environments. Studying the effects of natural selection on QTLs thus can be used to predict both the success or failure to establish in novel environments and the evolutionary potential for invasiveness in biological control organisms. These data could be invaluable in informing selective breeding programs for developing more effective biological control organism populations for subsequent introduction. Most methods to detect natural selection utilize diversity and polymorphism indices across the genome and are summarized in Table 1.

### 3.3. Gene Flow (Admixture/Migration)

Gene flow can occur to varying extents between proximal established populations of biological control organisms and even between established populations and newly introduced populations of biological control organisms.

Ongoing gene flow between newly introduced and established populations of biological control organisms [20,97,98,99,100,101] indicates the absence of environmental or reproductive barriers to hybridization. This process could indicate persistence and improved fitness of hybrids of colonizing and native populations through adaptive introgression [102,103]. Conversely, reduced or even no, contemporary gene flow could occur due to geographic or genomic barriers to migration. This process could signal the presence of population structure, inbreeding and reduced genomic diversity [104].

Beyond gene flow per se, reduced fitness of hybrid populations (outbreeding depression) has been observed during reintroduction episodes [105] due to epistasis between different genomic backgrounds. Estimating population structure and gene flow from genomic data can hence be used by biological control practitioners both to understand the successful establishment of newly introduced biological control organisms and to track genomic mechanisms of successful augmentation of previously established populations, both of which are otherwise intractable via observational studies.

### 3.4. Inbreeding

Non-random mating of close relatives in a population reduces genetic diversity, elevates homozygosity and fixes deleterious mutations (genetic load) [94,95,106]. This inbreeding depression not only reduces population fitness but also results in population structure due to genetic drift, wherein individuals within a subpopulation are genetically more similar to each other than to individuals from other subpopulations.

Inbreeding, although widely expected during primary introductions of species for biological control, is yet to be characterized in most species at the genetic level. Some cases of inbreeding have been reported in field populations of the convergent lady beetle, *Hippodamia convergens* [98] and in the Asian lady beetle, *Harmonia axyridis* [20]. However, understanding the long-term effects of inbreeding in these and other species using genomic data remains a nascent endeavor.

Estimating inbreeding using genetic data from populations of biological control organisms in conjunction with assays of fecundity, competition and efficiency of feeding on pests can inform success of biological control programs. For example, lab-inbred (Eastern and Western USA) populations compared to outbred (augmented Eastern-Western USA hybrid) populations of *H. convergens*, lack phenotypic variability despite genetic differences and exhibit equitable success in pea aphid utilization [107]. Tools to estimate inbreeding often use summary statistics such as Identity By Descent (IBD) probabilities, inbreeding coefficients and runs of homozygosity (ROH), often only delimited by the types of genetic data used to compute them.

## 4. Discussion and Recommendations

### 4.1. Genomic Considerations for Successful Biological Control

What comprises a successful biological control program? As summarized by [108] based on more than 800 studies, primary indicators of success in biological control are reduced pest abundance and increased pest mortality, relative target versus non-target effects and the type of biological control organism - generalist (polyphagous) versus specialist. In Table 2 we develop a population genomic framework for five measures of success of biological control organisms sensu [108]—(1) efficacy and establishment, measured using genetic diversity estimates; (2) spatio-temporal distribution, measured with divergence times and post-introductory evolutionary history; (3) managed breeding techniques, informed using studies of natural selection; (4) non-target effects and invasiveness, assessed via genetics of populations in imported or augmented environments; and (5) biotic effects on target/control organisms, measured using estimates of population structure, gene flow and inbreeding.

We propose that studies of success of biological control are essentially incomplete without a sufficient mix of manipulative experiments and genomics, which provide foundational insight into crucial ecological factors. Common denominators affect the ability of biological control organism populations to (i) establish, persist and grow in an introduced environment, (ii) withstand environmental and genomic pressures and evolve adaptively, (iii) avoid “escaping” into invasiveness and (iv), broadly, limit differential non-target effects. These factors are phenotypic differences in traits, which have underlying genomic differences within and between populations and ecological variation across geographically distinct populations of the species. Drawing on a classic example, the successful introduction of the vedalia beetle, *Rodolia cardinalis*, to suppress the cottony-cushion scale, *Icerya purchasi*, has been employed for over a century. Vedalia beetles are specialists, multivoltine, long-lived and highly efficient in obtaining prey [22]. Importantly, these are all ecological/phenotypic traits that can be characterized readily using genomic approaches [110,111]. Thus, experimental evolution and/or simulation studies based on existing genomic diversity of populations of the vedalia beetle (and other biological control organisms) could elucidate the effects of standing genomic variation on adaptability to novel environments. Efforts to quantify such variation in insect predators, including transcriptome and mitochondrial genome sequencing of *C. septempunctata* [112,113] and whole genome sequencing of *H. axyridis* (Havens et al., http://f1000research.com/posters/1096169), are underway. Additionally, a growing literature on landscape genomics methods (summarized in [114,115] highlights incorporating models of the distribution of populations in integrative studies of ecological and genomic variation [116]. Ultimately, new methods and software for jointly estimating demography and ecological parameters using genomic and geographical data should prove indispensable in studying the establishment of biological control organisms in novel environments.

### 4.2. Suggested Pipeline For Including Genomics Into Biological Control Programs

Rendering biological control more predictable, thus increasing the estimated 10 percent of reported attempts being successful, has been a long-term goal of applied ecologists and entomologists Gurr & Wratten, 1999 and Gurr et al., 2012 [2,13] argue that a majority of this failure rate concerns disregard for habitat requirements of the biological control organisms. They suggest that microhabitat manipulation (host ranges, prey/food availability, microclimates, etc.) ought to improve the chances of success. Although arguably true in several cases [117], genetic drift, natural selection and non-random mating surely play important and yet often undetermined roles as well [12,118,119]. However, predicting the evolutionary responses of organisms utilized in biological control is no easy task, as the number of contributing factors is formidable. Here we suggest four major population genetic processes—population size change, selection, gene flow and inbreeding—that, when quantified, can proffer important evidence of short- and long-term evolutionary trajectories of introduced organisms and their target species. Plummeting sequencing and genotyping costs and accelerated development of statistical methods and population genomics pipelines to estimate evolutionary parameters under a variety of demographic models, render these crucial insights more accessible. Thus, we propose a nine-step paradigm based on evolutionary and ecological principles—a ‘pipeline’ for applied ecologists and entomologists to enhance the likelihood of successful biological control programs.

Define biological questions about the system and build a hypothesized quantitative model of evolution based on mode of biological control. Is there a historical record of introductions in other regions, trophic-level interactions and ecological success parameters (described in Reference [13], including census size estimation and range expansion with host? For example, *H. axyridis* has successfully established populations across the world owing to importation for biological control and invasiveness. Due to its known historical record of introduction, Lombaert et al., 2010 [20] propose and test a model of hybridization of inbred Eastern and Western clusters of the species that putatively yielded the invasive Eastern North American population.Develop a sampling plan. Numerous studies [120,121] describe the issue of sample sizes, determined as(a) the number of individuals sampled per locale, (b) the number of sampling locales, (c) and the number and type of genomic loci analyzed. In short, although large sample sizes are preferable for estimating genomic diversity and differentiation, coalescent modeling and estimation of evolutionary history can work well with smaller sample sizes and greater number of genomic loci. Using replicated random samples of 3000 SNPs (Single Nucleotide Polymorphisms) from a large 2bRAD dataset from populations of the biological control organism *H. axyridis*, Li et al., 2020 [122] determined that a minimum of 6 individuals per population are sufficient to accurately estimate within- and between-population genomic diversity and differentiation. The ideal sampling plan should also be informed by the sequencing platform or protocol used for genotyping-by-sequencing, which is optimized to run up to 96 uniquely barcoded individuals to obtain thousands of informative sites.Conduct genotyping/sequencing. Strategies for obtaining molecular sequence or genotype information are contingent primarily on previously available genomic information from the species of interest. For example, many arthropod genomes are currently available (476 as of May 2020), with more in the works (see Arthropod Genomic Consortium, http://i5k.github.io/arthropod_genomes_at_ncbi) [123]. Alternatively, dense reduced representation library-based sequencing/genotyping [124] via technologies like RADseq [125] and PoolSeq [126] offer opportunities for demographic inference using SNPs in species with little prior genomic information. Meanwhile, repeat-based markers such as microsatellites continue to provide useful genetic insights into biological control organisms [20,21,98,127].Undertake preliminary bioinformatics steps involved in sequence/genotype clean-up, assembly, alignment and variant calling. Pipelines and tools have been developed to ease processing genomic/genotypic/sequencing data, including GATK [128], vcftools [129], SAMtools [130], BAMtools [131] and STACKS [132]. Resources for preliminary bioinformatics analyses are summarized under contributions of the Galaxy Project (www.galaxyproject.org) [133,134].Perform exploratory analyses. Calculate Method of Moments estimates of summary statistics, including heterozygosity, polymorphism, diversity indices, differentiation, allelic richness and inbreeding coefficients. Tools that bundle methods to estimate most basic summary statistics from genomic data include STACKS [132], VCFTools [129], PopGenome [135] and adegenet/pegas [136,137] packages in R (Table 3).Perform secondary analyses. Build data-sets (from whole genomic, reduced representation or genotypic data) that satisfy assumptions of the model or method of choice. Each method listed in Table 1 has its own set of caveats, assumptions and models, more details about which have been summarized in Reference [138].Simulate/estimate parameters under the model. The choice of programs for estimating demographic parameters depends on the type of genomic data (Table 1). Genotypic data (e.g., SNPs) are amenable for use in frequency-based statistics to infer demography and processes of divergent evolution. For instance, using SNP loci to compute divergence statistics (F_st_—[139] and other variants—[140,141], D statistic—ABBA-BABA tests—see References [60,142] can reveal migratory history between populations. Similarly, allele frequencies computed from individual loci can be used in likelihood and Bayesian methods to estimate population genetic structure and admixture, which is the basis of the widely cited program, STRUCTURE [41]. With ongoing improvements in sequencing technologies that offer high coverage and long reads, genotyping-by-sequencing technologies likely will be the go-to in terms of generating and analyzing large-scale population genomic data for biological control where no extensive whole genomic resources are available currently.Model selection. Demographic models often oversimplify the irrefutably complex reality of how populations evolve. However, statistics allow us to rigorously identify a model that explains the data better. Depending on the statistical methods applied, commonly utilized model-selection paradigms include likelihood ratio tests [54] and Akaike/Bayesian Information Criteria [143].Interpret estimated parameters under the “best” model, reconciling assumptions and biology of the system. The final step involves using a statistically informed explanation of the biological processes affecting populations of introduced biological control organisms and discussing the caveats of using model-based population genomics.

## 5. Conclusions

Beginning with the development of biological control as a major tactic for pest management during the 20th century, an appreciation that biological control was not only applied ecology but also had a foundation in genetics and evolution, was gained. Still, for most of the 1900s, the major emphasis of the discipline remained on ecological principles, with notable exceptions [152,153,154]. During the past 25 years, as molecular tools have been applied to address evolutionary questions in biological control, we have gained a deeper appreciation of transgenerational processes. Emerging topics examined in relation to biological control include manipulating genetic variation in biological control organisms [155], using molecular tools in importation biological control [156], revealing microevolution [17] and examining evolutionary concepts in importation biological control [157,158]. In this spirit, Evolutionary Applications dedicated an issue to focus on evolution and biological control [159]. Within this scholarly work, an appreciation of the influence of new cutting-edge tools on the discipline was recognized. For example, Roderick et al. [159] identified next-generation sequencing, computational modeling and bioinformatics as approaches that would enhance our understanding of evolution in biological control. In our review, we specifically focus on harnessing the power of population genomics, including next-generation sequencing and demographic modeling, to provide a more predictive basis and evolutionary understanding for biological control. With the rapid development and application of sophisticated molecular and computational tools and approaches, we show how new perspectives and insights can be gained on long-standing questions related to the genetic bases and evolutionary outcomes of human manipulation of biological control organisms for the management of pest species.

## Figures and Tables

**Figure 1 insects-11-00462-f001:**
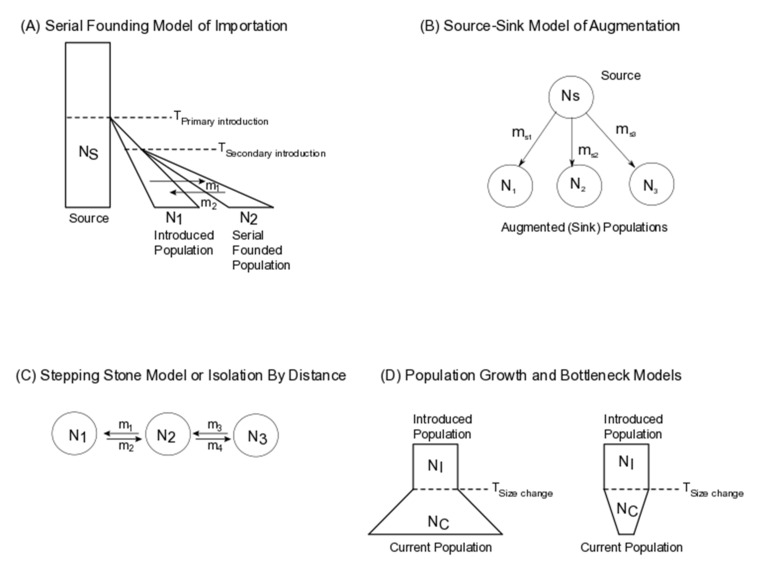
Population genetic models that are used to describe importation and augmentation of biological control organisms—(**A**) Serial founder model, often used to describe importation of biological control organisms, (**B**) Source-Sink model to describe augmentation, (**C**) Stepping stone model to describe establishment of new populations post-importation or augmentation, and (**D**) Population Growth and Bottleneck models to describe successful establishment or failure of importation and augmentation.

**Table 1 insects-11-00462-t001:** List of commonly used population genomics tools for estimating evolutionary history under a variety of models.

Software	Statistical Method	Citation	Purpose	Availability
STRUCTURE	Bayesian MCMC	Pritchard et al., 2000 [41]	Estimating admixture proportions, ancestral subpopulation allele frequencies.	OS, Binaries
PSMIX	ML	Wu et al., 2006 [42]	Estimating admixture proportions, ancestral subpopulation allele frequencies.	OS, R package
ADMIXTURE	ML	Alexander et al., 2009 [43]	Estimating admixture proportions, ancestral subpopulation allele frequencies.	Binary only
FRAPPE	ML	Tang et al., 2005 [44]	Estimating admixture proportions, ancestral subpopulation allele frequencies.	Binary only
EIGENSTRAT	PCA	Price et al., 2006 [45]	Estimating population stratification	OS, Binaries
IM	Bayesian MCMC	Hey and Nielsen 2004 [46]	Estimating ancestral demography under an Isolation with migration model	OS, Binaries
IMa2	Bayesian MCMC	Hey and Nielsen 2007 [47], Hey 2010 [48]	Estimating ancestral demography under an Isolation with migration model	OS, Binaries
IMa2p	Bayesian MCMC	Sethuraman and Hey 2016 [49]	Estimating ancestral demography under an Isolation with migration model	OS
MIGRATE	Bayesian MCMC	Beerli and Felsenstein 2001 [50], 1999 [51], Beerli 2008 [52]	Estimating ancestral demography under an island model	OS, Binaries
BayesAss	Bayesian MCMC	Wilson and Rannala 2003 [53]	Estimating recent migration under a divergence model	OS, Binaries
MDIV	Bayesian MCMC	Nielsen and Wakeley 2001 [54]	Estimating ancestral demography under an Isolation with migration model	OS, Binaries
LAMARC	Bayesian MCMC	Kuhner 2006 [55]	Estimating ancestral demography under an island model	OS, Binaries
DIYABC	ABC	Cornuet et al., 2010 [56]	Testing complex population histories and estimate parameters	OS, Binaries
MSVAR	Bayesian MCMC	Beaumont 2003 [57]	Estimating population size change under a panmictic model	OS
FASTRUCT	ML	Chen et al., 2006 [58]	Estimating admixture proportions, ancestral subpopulation allele frequencies.	Binary only
BAPS	Bayesian MCMC	Corander et al., 2006 [59]	Estimating admixture proportions, ancestral subpopulation allele frequencies.	Binaries only
ADMIXTOOLS	Summary Statistics	Patterson et al., 2012 [60]	Tests of admixture occurrence	OS
TREEMIX	ML	Pickrell and Pritchard 2012 [61]	Inferring divergence and mixtures from genomic data	OS
FLUCTUATE	Bayesian MCMC	Kuhner, Yamato and Felsenstein 1998 [62]	Inferring population size change from genetic data	OS
BOTTLENECK	Bayesian MCMC	Cornuet and Luikart 1996 [40]	Inferring population size bottlenecks from genetic data	Binary only
FASTRUCTURE	Bayesian MCMC	Raj et al., 2014 [63]	Inferring population structure from SNP data	OS
GPHOCS	Bayesian MCMC	Gronau et al., 2012 [64]	Inferring demography from individual genome sequences	OS
PSMC	HMM	Li and Durbin 2010 [65]	Inferring population size history from diploid genomes	OS
FASTSIMCOAL2	Bayesian MCMC, ML	Excoffier et al., 2013 [66]	Inferring ancestral demography from SNP data	Binary only
DADI	ML	Gutenkunst et al., 2010 [67]	Inferring ancestral demography from SNP data, testing complex population histories	OS
ABCreg	ABC	Excoffier et al., 2009 [68]	Testing complex population histories and estimate parameters	OS
STRUCTURAMA	Bayesian MCMC	Huelsenbeck and Andolfato 2011 [69]	Estimating admixture proportions, ancestral subpopulation allele frequencies.	OS
DICAL	HMM	Sheehan et al., 2013 [70]	Inferring demography from individual genome sequences	OS
SWEED	ML, LLR	Pavlidis et al., 2013 [71]	Inferring selective sweeps	OS
SWEEPFINDER2	ML, LLR	DeGiorgio et al., 2016 [72]	Inferring selective sweeps	OS
MLNE	ML	Wang and Whitlock 2003 [73]	Inferring contemporary effective population size	OS
LDNE	Summary Statistics	Do et al., 2014 [74]	Inferring contemporary effective population size	Binary only

ML = Maximum Likelihood, MCMC = Markov Chain Monte Carlo, LLR = Likelihood Ratio Test, PCA = Principal Components Analysis, OS = Open Source.

**Table 2 insects-11-00462-t002:** Indicators of success of biological control programs and how we can measure/estimate these using population genomic methods. All methods listed either utilize microsatellite or Short Tandem Repeat (STR) markers, Single Nucleotide Polymorphisms (SNPs) or haplotype data generated from common genotyping and sequencing platforms.

Category	Ecological Parameters	Evolutionary Parameters	Genomic Method	Evolutionary Perspective
Agent efficacy, establishment	Mortality/survivorship, abundance before/after release	Effective population size	Contemporary Ne—Colony2, ONeSamp, Estim, etc.—see Gilbert and Whitlock 2015 [109], Ancestral and current Ne—IM, IMa2, IMa2p, MIGRATE, LAMARC, PSMC	Ne measures the size of the natural enemy population evolving neutrally by genetic drift. It differs from census sizes, in that it offers a perspective on genetic diversity and hence adaptability of the population, response to new environments and resilience to failed introductions. Ancestral Ne versus current Ne thus determines increase or decrease in genomic diversity.
		Diversity, polymorphism, heterozygosity, homozygosity, differentiation, inbreeding coefficients	Genepop, Arlequin, ADEGENET, DNASP, MEGA	Broadly lumped together as genomic diversity indices, all these indices are indicators of the ’genetic health’ of the introduced population. Successful control programs would thus expect sustainable natural enemy populations to have higher genetic diversity, polymorphism, differentiation with respect to other populations and thus lower homozygosity and inbreeding.
Spatio-temporal distribution	Spatial, temporal scale assessment of abundance, distribution	Divergence times, time since population size change, phylogeography	TreeMix, IM, IMa2, IMa2p, BEAST, DIYABC, MrBayes, Bottleneck, MSVAR, FLUCTUATE, LAMARC, GeoPhyloBuilder, etc.	Divergence time estimates provide evidence of time since introduction of natural enemies. Similarly, time since population size change can be used to estimate times of bottlenecks or invasiveness. Phylogeography studies also allow overlaying the current phylogenetic tree over geographical data.
Agent management techniques	Agent manipulation by strain selection	Selection, demography	Fst-GWAS, SweepFinder, SweeD, McDonald-Kreitman tests	Estimating genome-wide selection across strains allows prediction of genotype-phenotype interactions and efficacy of selection in adaptive evolution of the natural enemy population to be introduced.
Non-target effects, invasiveness	Other species, other than target/pests	Selection, demography		Ancestral and current demography, genomic diversity, differentiation and inbreeding coefficients can be used as a proxy for competition or predation of non-target species or populations.
			QTL mapping	Understanding underlying traits of adaptive evolution and invasiveness.
Biotic effects on target/agents	Inter-, intra-guild predation, competition	Admixture, migration, inbreeding	Admixture—STRUCTURE, Admixture, MULTICLUST, BAPS, TREEMIX Migration—MIGRATE, LAMARC, IMa2, IMa2p, IM, GPhoCS, DIYABC	Admixture (and migration) between stock and native populations is a measure of degree of hybrid compatibility and increase in genomic diversity due to gene flow. Similarly, lack thereof is a measure of predation/competition and genome-level incompatibilities. Successful biological control populations would thus be expected to have higher levels of admixture and bidirectional migration with local populations (especially in augmentative bio-control).

**Table 3 insects-11-00462-t003:** List of commonly used tools/pipelines for preliminary analyses (data compilation, assembly, filtering, quality control, formatting) of population genomic data.

Software	Citation	Type of Data	Purpose
VCFTOOLS	Danecek et al., 2011 [129]	Genomic, SNP	Variant calling, summary statistics, data filtering, file manipulation
SAMTOOLS	Li et al., 2009 [130]	Genomic, multiple sequence alignment	Data filtering, cleanup, multiple sequence alignment, file manipulation
BAMTOOLS	Barnett et al., 2011 [131]	Genomic, multiple sequence alignment	Data filtering, cleanup, multiple sequence alignment, file manipulation
GATK	McKenna et al., 2010 [128]	Genomic, SNP	Variant calling, summary statistics, data filtering
GALAXY PROJECT	Blankenberg et al., 2010 [134]	All	Suite of pipelines for numerous bioinformatics analyses of genomic data
JVARKIT	Lindenbaum 2015 [144]	Genomic, SNP	Suite of tools for data filtering, file manipulation, cleanup
SNP-SITES	Page et al., 2016 [145]	Genomic, SNP	Variant calling
BIOCONDUCTOR	Gentleman et al., 2004 [146]	All	Suite of pipelines for numerous bioinformatics analyses of genomic data
ADEGENET/PEGAS	Jombart 2008 [136], Paradis 2010 [137]	Genomic, SNP	Suite of tools for data filtering, file manipulation, cleanup
POPGENOME	Pfeifer et al., 2014 [135]	Genomic, multiple sequence alignment	Suite of tools for data filtering, file manipulation, cleanup
STACKS	Catchen et al., 2011 [132]	RAD, SNP	Variant calling, summary statistics, data filtering, file manipulation
MEGA6	Tamura et al., 2013 [147]	Multiple sequence alignment, microsatellite, SNP	Summary statistics
GENEPOP	Rousset 2002 [148]	Multiple sequence alignment, microsatellite, SNP	Summary statistics
ARLEQUIN	Excoffier et al., 2010 [149]	Multiple sequence alignment, microsatellite, SNP	Summary statistics
DNASP	Librado and Rozas 2009 [150]	Multiple sequence alignment, microsatellite, SNP	Summary statistics
BEDTOOLS	Quinlan 2014 [151]	Genomic, SNP	Data filtering, cleanup, multiple sequence alignment, file manipulation

RAD = Restriction Associated Digestion, SNP = Single Nucleotide Polymorphism (also called variants).

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
