# Peer review of "Insights from Population Genomics to Enhance and Sustain Biological Control of Insect Pests"

_insects, 2020, doi:10.3390/insects11080462_

Round 1

Reviewer 1 Report

The review is very interesting and full of classical definitions of populations genetics, I have added minor comments and suggestion on the file attached. I would recommend to explore more the genomic data acquisition, at least example in the table. I do recommend for publication in Insects after minor revisions.

All the best.  

Author Response

Reviewer 1

The review is very interesting and full of classical definitions of populations genetics, I have added minor comments and suggestion on the file attached. I would recommend to explore more the genomic data acquisition, at least example in the table. I do recommend for publication in Insects after minor revisions.

We thank the reviewer for all their insightful comments in the manuscript PDF – we have now addressed all these comments and tracked changes in MS Word so they can be easily accessed.

Reviewer 2 Report

Biological control is used as an ecologically friendly way of pest management against pest insects since more than 100 years without exact knowledge on the ecological consequences of releasing predators or parasitoids.With the use of moleculat tools and sophisticated statistical methods it became possible to understand the evolutionary background of biocontrol.

Authors here present a mainly theoretical overview on studying population structures in biocontrol, but not without mentioning examples of the respective Approach (mainly from North America). 

The review is logically structured and ends with a nine-step pipeline for  successful use of biocontrol by applied ecologists and entomologists.

The manuscript is generally well written and needs only minor editorial improvement:

  • if a sentence begins with a parenthesis, it must also end with one, even when there is a reference in square brackets
  •  References have to be carefully checked for a uniform style of presentation according to Insects instructions for authors

Author Response

Reviewer 2

Biological control is used as an ecologically friendly way of pest management against pest insects since more than 100 years without exact knowledge on the ecological consequences of releasing predators or parasitoids.With the use of moleculat tools and sophisticated statistical methods it became possible to understand the evolutionary background of biocontrol.

Authors here present a mainly theoretical overview on studying population structures in biocontrol, but not without mentioning examples of the respective Approach (mainly from North America). 

The review is logically structured and ends with a nine-step pipeline for  successful use of biocontrol by applied ecologists and entomologists.

The manuscript is generally well written and needs only minor editorial improvement:

  • if a sentence begins with a parenthesis, it must also end with one, even when there is a reference in square brackets
  • References have to be carefully checked for a uniform style of presentation according to Insects instructions for authors

We have now addressed all the parentheses issues (as also pointed out by Reviewer 3). We hope that the reference style is now uniform as well across the manuscript.

Reviewer 3 Report

The manuscript entitled “Insights from Population Genomics to Enhance and Sustain Biological Control of Insect Pests” is well written and reports an extensive overview of the possible application  of genomic studies on population dynamic and fate of BCAs. The manuscript requires some minor revisions. All the initial part is quite general. In this section, the style and organisation are a bit more adapted to a text book for University students than a review. For example, box 1 it’s a bit didactic.  In all the manuscript the use of brackets is unusual. There also several typos (e.g. line 385)

The review focus on arthropod BCAs. I suggest to clearly specify early in the text. BCAs includes also microorganisms and other invertebrates (e.g. entomophagus nematodes).

Line 42. I did not understand what is b) and d). Actually there are more ecological interaction that have not been mentioned. For example, BCAs may be preys of birds or other animal or may be vector of detrimental pathogens

Line 54. Delete e.g. (

Line 107. Write the Authors name instead of [26] and cite [26] at the end of the sentence

Line 134 ) missing

Line 135 (missing

Line 138 delete )

Line 145 ) missing

Line 147 (missing

Line 213-14. Sentence could be deleted

Line 224-per se, italics

Author Response

Reviewer 3:

 The manuscript entitled “Insights from Population Genomics to Enhance and Sustain Biological Control of Insect Pests” is well written and reports an extensive overview of the possible application of genomic studies on population dynamic and fate of BCAs. The manuscript requires some minor revisions. All the initial part is quite general. In this section, the style and organisation are a bit more adapted to a text book for University students than a review. For example, box 1 it’s a bit didactic. In all the manuscript the use of brackets is unusual. There also several typos (e.g. line 385)

The review focus on arthropod BCAs. I suggest to clearly specify early in the text. BCAs includes also microorganisms and other invertebrates (e.g. entomophagus nematodes).

We have now added this at the beginning of the review.

Line 42. I did not understand what is b) and d). Actually there are more ecological interaction that have not been mentioned. For example, BCAs may be preys of birds or other animal or may be vector of detrimental pathogens

We have now edited this to be more explanatory, and also added how microbial pathogens can be detrimental to other organisms that interact with BCA’s.

Line 54. Delete e.g. (

Line 107. Write the Authors name instead of [26] and cite [26] at the end of the sentence

Line 134 ) missing

Line 135 (missing

Line 138 delete )

Line 145 ) missing

Line 147 (missing

Line 213-14. Sentence could be deleted

Line 224-per se, italics

We have now corrected all typos in the manuscript, as pointed out by this reviewer. Additionally, we hope that both Box 1 and the introduction section will provide a much needed review of population genomics concepts to the “Insects” readership